Intraspecific phenotypic variation in life history traits of Daphnia galeata populations in response to fish kairomones

Tams Verena
Lüneburg Jennifer
Seddar Laura
Detampel Jan-Phillip
http://orcid.org/0000-0001-7376-4560 Cordellier Mathilde mathilde.cordellier@uni-hamburg.de
Institut für Zoologie, Universität Hamburg , Hamburg , Germany
Forbes Valery
Electronic publication date: 2018 Oct 17
Publication date: 2018
Volume: 6
Electronic Location ID: e5746
Received 2018 Mar 8; Accepted 2018 Sep 14
Copyright: © 2018 Tams et al.
Copyright year: 2018
Copyright holder: Tams et al.
License: This is an open access article distributed under the terms of the Creative Commons Attribution License, which permits unrestricted use, distribution, reproduction and adaptation in any medium and for any purpose provided that it is properly attributed. For attribution, the original author(s), title, publication source (PeerJ) and either DOI or URL of the article must be cited.
License URL: https://creativecommons.org/licenses/by/4.0/

Keywords: Phenotypic plasticity, Life history traits, Population ecology, Predator-induced response, Daphnia, Adaptive potential

Funding: The Volkswagen Foundation 86030 This work was supported by the Volkswagen Foundation (Grant No. 86030). The funders had no role in study design, data collection and analysis, decision to publish, or preparation of the manuscript.

==============================
Phenotypic plasticity is the ability of a genotype to produce different phenotypes depending on the environment. It has an influence on the adaptive potential to environmental change and the capability to adapt locally. Adaptation to environmental change happens at the population level, thereby contributing to genotypic and phenotypic variation within a species. Predation is an important ecological factor structuring communities and maintaining species diversity. Prey developed different strategies to reduce their vulnerability to predators by changing their behaviour, their morphology or their life history. Predator-induced life history responses in Daphnia have been investigated for decades, but intra-and inter-population variability was rarely addressed explicitly. We addressed this issue by conducting a common garden experiment with 24 clonal lines of European Daphnia galeata originating from four populations, each represented by six clonal lines. We recorded life history traits in the absence and presence of fish kairomones. Additionally, we looked at the shape of experimental individuals by conducting a geometric morphometric analysis, thus assessing predator-induced morphometric changes. Our data revealed high intraspecific phenotypic variation within and between four D. galeata populations, the potential to locally adapt to a vertebrate predator regime as well as an effect of the fish kairomones on morphology of D. galeata.

Introduction

Intraspecific phenotypic variation is crucial for the persistence of a population, since low intra-population variation increases the risk of extinction (Bolnick et al., 2011; Scheiner & Holt, 2012; Forsman, 2014). Loss of phenotypic variation can be caused by the reduction of genetic variation, for example, due to genetic drift (random loss of alleles) (Vanoverbeke & De Meester, 2010; Bolnick et al., 2011), inbreeding depression (Lynch, 1991; Swillen, Vanoverbeke & De Meester, 2015) or positive selection (Biswas & Akey, 2006). On the contrary, phenotypic variation can increase as a consequence of environmental change (biotic and/or abiotic) as well as through an increase in genetic variation, which in turn occurs through gene flow (migration), mutation and recombination (Griffiths et al., 2000). Phenotypic variation ‘is the fuel that feeds evolutionary change’ because natural selection acts on it (Stearns, 1989). Phenotypic plasticity describes the ability of genotypes to produce different phenotypes depending on the environment, helping organisms to survive and reproduce in heterogeneous environment (Stearns, 1989; Agrawal, 2001). Phenotypic plasticity implies an adaptive potential to locally adapt to a changed environment (Stearns, 1989). If the phenotypically plastic organism produces a modified and successful phenotype whose fitness (higher reproductive success) is higher than an unmodified phenotype, the underlying genotype contributes more to the genetic make-up of the whole population.

Predation structures whole communities (Werner & Peacor, 2003; Beschta & Ripple, 2009; Boaden & Kingsford, 2015; Aldana et al., 2016), drives natural selection within populations (Morgans & Ord, 2013; Kuchta & Svensson, 2014) and maintains species diversity (Estes et al., 2011; Fine, 2015). Aquatic predators release chemical substances, so called kairomones, into the surrounding waters which can be detected by their prey. Both vertebrates (Schoeppner & Relyea, 2009; Stibor, 1992) and invertebrates (Machacek, 1991; Stibor & Lüning, 1994) release kairomones, triggering specific phenotypic plastic responses such as morphological or behavioural changes (Dodson, 1989; Schoeppner & Relyea, 2009). The predator-induced defences can be highly variable within a species, depending on factors such as the predator and colonization histories (Eklöv & Svanbäck, 2006; Kishida, Trussell & Nishimura, 2007; Edgell & Neufeld, 2008).

Invertebrate as well as vertebrate predator kairomones have been shown to cause phenotypic plastic responses in Daphnia. These induced responses are predator specific and vary across Daphnia species. Behavioural changes such as diel vertical migration (Effertz & Von Elert, 2015) and the associated metabolic costs (Dawidowicz & Loose, 1992), aggregation and escape response (Pijanowska, 1997), reduction of ingestion and filtration rates (Rose, Warne & Lim, 2003; Beckerman, Wieski & Baird, 2007; Pestana, Baird & Soares, 2013), depth selection (Cousyn et al., 2001), increased alertness (Boersma, Spaak & De Meester, 1998) and diapause (production of resting eggs = ephippia) (Pijanowska & Stolpe, 1996) were reported for different daphnids exposed to vertebrate predator kairomones (fish). Diverse morphological changes have been shown to occur in the presence of kairomones of the invertebrate predator Chaoborus, such as the production of neck teeth in D. pulex (Lüning, 1995; Tollrian, 1995) or the famous helmets of D. longispina (Brett, 1992) and D. cucullata (Agrawal, Laforsch & Tollrian, 1999). Recently Herzog et al. (2016) observed a remarkable morphological change of D. barbata exposed to Triops kairomones. D. barbata changes its whole body symmetry to an S-shape, presumably to impede ingestion by their invertebrate predator. Apart from morphology, physiology and behaviour, predator kairomones were also shown to influence life history traits in different Daphnia species. Among others, size and fecundity, two important traits for population survival, were affected, resulting in earlier maturation (Riessen, 1999; De Meester & Weider, 1999; Weber, 2003; Castro, Consciência & Gonçalves, 2007) and smaller size (Stibor & Lüning, 1994; Castro, Consciência & Gonçalves, 2007). Size is a very important factor for survival in the face of fish predation, since small individuals are more likely to go undetected. These predator-induced responses are the result of phenotypic plasticity and their magnitude might play a role in adaptation.

The ability of Daphnia to locally adapt to different stressors has been demonstrated before, for example, for fish as a vertebrate predator (Boersma, Spaak & De Meester, 1998; Cousyn et al., 2001; Declerck & Weber, 2003) and pesticides (Jansen et al., 2011). Differences in phenotypic responses between Daphnia populations can further happen because of the high genetic divergence between lakes, which is turn due to specific colonization patterns in Daphnia (De Meester et al., 2002), monopolization effects, and overall lack of gene flow. Hence, we expect the investigated populations to exhibit population specific responses. Clonal variation of Daphnia has been regularly reported (Machacek, 1991; Castro, Consciência & Gonçalves, 2007; Beckerman, Rodgers & Dennis, 2010). Some investigators evaluated clonal variation within one population (Cousyn et al., 2001; De Meester & Weider, 1999) or among populations of Daphnia (Boersma, Spaak & De Meester, 1998; Declerck & Weber, 2003; Boeing, Ramcharan & Riessen, 2006; Hamrová, Mergeay & Petrusek, 2011; Lind et al., 2015). Others rarely used more than one or two clonal lines per population, drawing conclusions based on single clonal lines. Although intra-population variation or lack thereof is relevant to population maintenance in the face of predation pressure, the relative importance of the intra- and inter-population variation was rarely measured. We are aware of only a few studies which addressed this aspect of predator-induced responses in Daphnia. Boersma, Spaak & De Meester (1998) used four clonal lines of D. magna for each of the four population showing that the strength and combination of responsive traits can differ across genotypes (clonal lines). The clonal variation in D. pulex exposed to vertebrate (fish) and invertebrate (Chaoborus) predator kairomones was assessed for migration behaviour (Boeing, Ramcharan & Riessen, 2006) and life history traits (Lind et al., 2015) at the inter- and intra-population level. Recently, Reger et al. (2018) showed that predation drives local adaptation in phenotypic plasticity in 70 clonal lines of D. pulex.

In the present study, we assess the intraspecific phenotypic variation of European D. galeata in the presence of fish kairomones, by measuring shifts in life history traits as well as morphological changes in a total of 24 clonal lines (six clonal lines within each of the four populations). We expect that (i) there is intraspecific phenotypic variation, evidenced by inter-clonal variation within as well as among populations. We hypothesize that (ii) local adaptation and population divergence caused by drift and limited gene flow shape the observed phenotypic variation, and is evidenced by population specific responses. Finally, we expect that (iii) the exposure to fish kairomones affects the morphology in D. galeata. We hypothesize that (iv) a correlation between life history change and morphology exists. Specifically, we hypothesize that females which increased their total number of offspring in the presence of fish kairomones, change their morphology towards a bulkier shape to accommodate more eggs.

Materials and Methods

Experimental organisms and lakes of origin

This study integrated 24 D. galeata clonal lines from four different locations: Lake Constance (popLC), Germany; Greifensee (popG), Switzerland; Müggelsee (popM), Germany and Jordan Reservoir (popJ), Czech Republic. These are all permanent lakes with a large water body and varying densities of planktivorous fish (Table 1).

Table 1 Background information of ecological aspects of the four European lakes of which experimental clonal lines originate from.

Lake	Greifensee	Jordan Reservoir	Lake Constance	Müggelsee	
Abbreviation	popG	popJ	popLC	popM	
Location	Switzerland	Czech Republic	Austria, Germany, Switzerland	Germany	
GPS coordinates	47°21′20″N, 8°40′10″E	49°24′55″N, 14°39′49″E	47°37′21″N, 9°26′24″E	52°26′6″N, 13°38′6″E	
N	6	6	6	6	
Alt. (m)	435	437	395	32	
Vol. (km3)	0.1485	0.0027	48	0.0366	
Dep. (m)	34	14	254	8	
Av. Dep. (m)	18	4.5	90	4.9	
Stratification	Dimictic	Dimictic	Monomictic	Polymictic	
Fish biomass (kg/ha)	32	607.5	54	70–100	
Presence of Leuciscus sp.	Yes	Yes	Yes	Yes	
Note:

N, Number of genotypes; Alt., Altitude; Vol., Volume; Dep., Maximum depth; Av. Dep., Average depth.

Clonal lines were established from dormant eggs from sediment cores and have been used in previous studies (Henning-Lucass et al., 2016; Herrmann et al., 2017). These ephippia originate from sediment layers of the recent decade (2000–2010) and were subjected to hatching stimuli described in detail in Henning-Lucass et al. (2016). The clonal lines were maintained in lab cultures (18 °C, 16 h light/8 h dark cycle, food: Acutodesmus obliquus, medium: Aachener Daphnien Medium (ADaM) (Klüttgen et al., 1994)) for up to 5 years and no less than 3 years prior to the present experiment.

Fish densities and hence predation pressures are not easy to measure very precisely due to seasonal variation (Fischer & Eckmann, 1997), variation between years (Kruuk, Conroy & Moorhouse, 1991) as well as differences between lakes (Mehner et al., 2005). We provide here estimates of fish biomass available from the literature corresponding as closely as possible to the time period the resting eggs came from. The highest estimate of fish biomass was for the Jordan Reservoir, which is stocked for recreational purposes and the lowest for Greifensee (Table 1). Fish belonging to the Leuciscus genus were found in all lakes from which our clonal lines originate (Philipp, 2006; Kubecka & Bohm, 1991; Alexander et al., 2016; Fischereiamt Berlin, 2013).

Media preparation

The basic medium was ADaM for fish and Daphnia cultures. Two types of media were used for breeding and experimental conditions: fish kairomone and control medium. In total forty ide (Leuciscus idus) were maintained in an aerated, separate 200 L aquarium, in which they were fed frozen Daphnia cubes and dry food. The ide or closely related species are present in all the studied lakes (Table 1). Previous studies showed that ide elicit plastic responses in D. galeata clonal lines from Lake Constance (Sakwinska, 2002) and Greifensee (Wolinska, Löffler & Spaak, 2007). Fish medium was obtained by keeping five randomly chosen ide in an aerated 20 L aquarium for 24 h to produce fish kairomone medium. The fish were not fed in the fish medium production tank to avoid Daphnia alarm cues to be mixed with the fish kairomones. The fish kairomone media imitates a scenario of high fish density (Cousyn et al., 2001; Swillen, Vanoverbeke & De Meester, 2015). Control medium was produced in an aerated, separated aquarium and handled first, before handling of fish and fish medium. All media was filtered before use to remove faeces from predators and bacteria larger than 1.2 μm (Whatman, membrane filters, ME28, Mixed cellulose-ester, 1.2 μm). All media were supplemented with 1.0 mg C L-1, P rich Acutodesmus obliquus before use and exchanged daily (1:2) to guarantee a nutrient rich environment and a constant fish kairomone concentration. The algae concentration was calculated from photometric measurement of the absorbance rate at 800 nm.

Because fish was used to produce fish kairomone media, this experiment was subject to approval through the ‘Behörde für Gesundheit und Verbraucherschutz’ of the City of Hamburg. It was approved under the number 75/15.

Experimental design and procedures: life table experiment

Prior to the experiment, each clonal line was bred in kairomone-free water (control environment) and in kairomone water (fish environment) for two subsequent generations to minimize inter-individual variances. To this end, 10–15 egg-bearing females per clonal line were randomly selected from mass cultures. From these females of unknown age, neonates were collected and raised under experimental conditions and served as grandmothers (F0) for the experimental animals (F2). Neonates of the third to fifth brood carried by the F0 animals were used as breeding (F1) animals. Neonates of the third to fifth brood carried by the F1 animals were used in turn as experimental individuals (F2). A pair of neonates was introduced in the experimental vessels (50 mL glass tube) at the start of the experiment to compensate for eventual mortality. One of the individuals was randomly discarded when necessary at day 4 (t4), so that one individual remained in each vessel. This procedure was applied to F1 and F2 individuals. Fifteen replicates were used per environment and per genotype (clonal line). Sister neonates of F2 (n = 15) were collected in 70% ethanol for size measurements at day 0 (t0). Life history parameters were recorded daily during the experiment. Before media renewal, females were checked for maturation and neonates were counted, removed and preserved in ethanol every day. Adults were preserved in ethanol as well at the end of the experiment. The experiment lasted for 14 days (t14) for each experimental individual to monitor the performance of each clonal line within a fixed period of time.

Cetyl alcohol was used to break the surface tension of the media during breeding and the experiment to reduce juvenile mortality (Desmarais, 1997). Breeding and experimental phases were conducted at a temperature of 20 °C and a 16 h light/8 h dark cycle in a brood chamber with a light intensity of 30% (Rumed, Type 3201D).

The experiment was conducted in three experimental rounds due to logistic reasons, and clonal lines from all four populations were included in each round (Table S1). Previous pilot studies showed that ensuring synchronicity of so many clonal lines at once were extremely difficult.

Data collection and analysis

Life history traits

Life history parameters such as age at first reproduction (‘AFR’) [d], number of neonates per brood per female, total number of broods per female (‘broods’), total numbers of neonates per female (‘offspring’), size of first clutch (‘brood1’) [number of neonates per female], ‘survival’ [%] and somatic growth rate (‘SGR’) [μm d−1] were recorded. ‘AFR’ was the day of releasing the first brood from the brood pouch, with neonates swimming in the vessel. For further analysis the average value of the 15 individuals per clonal line (genotype) per environment (treatment) was calculated for each life history trait to estimate the clonal response to a kairomone (fish) vs. kairomone-free (control) environment. Survival rate was defined as the proportion of females surviving from the day of separation (t4) until the end of the experiment (t14). Reproductive rate was calculated by dividing the total number of offspring per female by the total number of broods per female. Relative fitness (w) was calculated by multiplying survival and reproductive rate of a clonal line before dividing by the maximum survival and reproductive rate of the other clonal lines within population (‘relnest’) and among all populations (‘relclone’). Some clonal lines produced male offspring during breeding and the experiment. Males occurred at very low frequencies and were excluded from the data analysis. We aimed to test a total of 720 individuals in this experiment (24 clonal lines × 2 environments × 15 replicates). In total we measured life history traits for 684 experimental individuals (Table S1).

Digitizing of experimental animals for ‘size’ and ‘shape’ analysis

Digital photographs of Daphnia preserved in ethanol were taken with a stereomicroscope (Nikon SMZ800N) at a magnification of 60× for neonates (t0) and 40× for adults (t14) with NIS-elements 4.3 software. All experimental individuals were photographed in lateral view (left body side up).

Measurement of body length (‘size’)

Body length was measured from the top of the head through the middle of the eye to the ventral basis of the spine, excluding the spine itself. ‘SGR’ (μm/day) was calculated by subtracting the average length of neonates at the beginning of the experiment (t0; n = 15) from the length of each adult individual at the end of the experiment (t14), divided by the complete experimental time in days. The measurement error of digitizing and measuring the length of the same individual 10 times was +/−3.24 μm (SD). The measurement error of measuring 10 times the length of an individual using the exact same picture was +/−1.67 μm (SD).

Geometric morphometric analysis of the ‘shape’ of the body

Since the morphology of Daphnia does not allow the assignment of many landmarks, we decided to use the semilandmark approach. Semilandmarks are a set of individual landmarks which are interpolated to represent the curve of a structure (Zelditch et al., 2004). Landmarks and semilandmarks were assigned on a subset of digital images of adult experimental individuals (max. n = 10 per clonal line and environment, with a total of 459 individuals; Table S1) according to Zelditch et al. (2004). In total three landmarks and 115 semilandmarks were assigned on each individual photograph. The first landmark was appointed to the tip of the rostrum, the second in the middle of the eye and the third at the ventral basis of the spine. In our study the first curve consisted of 70 interpolated landmarks (=semilandmarks) along the dorsal body outline, starting at the first landmark and ending on the dorsal basis of the spine. The second set of semilandmarks consisted of 45 semilandmarks along the ventral body outline, starting at landmark three and ending opposite of the dorsal basis of antenna. After the assignment of landmarks and semilandmarks, X and Y coordinates were recorded using TpsDig2 (Rohlf, 2015). A General Procrustes analysis (GPA) was performed using the package ‘geomorph’ in R (Adams & Otárola-Castillo, 2013). The measurement variance for assigning landmarks and semilandmarks of an individual using the exact same picture was <0.0001. Investigators of ‘shape’ measurements worked with a blind data set, not knowing which individual belongs to which group (environment, genotype and population).

Statistical analysis

All statistical analyses for life history traits were performed and all figures were created using R version 3.3.1 (R Core Team, 2016). For the generalized linear mixed models the package ‘lme4’ was used (Bates et al., 2015). Visualization of life history traits were performed by using the package ‘ggplot2’ (Wickham, 2016). For the geometric morphometric analysis the package ‘geomorph’ was used (Adams & Otárola-Castillo, 2013). The visualization of shape differences was performed with the R package ‘shapes’ (Dryden, 2017). R scripts are provided in Supplementary Materials.

To compare life history traits between the different populations in the presence and absence of fish kairomones, we applied generalized linear mixed effect models for each trait, except ‘shape.’ Visual inspection of residual plots as well as the Shapiro–Wilk-test revealed deviations from homoscedasticity for each trait, supporting the decision to use nonparametric models for statistical analysis. Hence, error distributions were assigned individually per trait. We used ‘Environment × Population’ and ‘Genotype × Environment’ as fixed categorical factors in our models. To account for genotype differences among populations, we included ‘Clone’ nested within ‘Population’ as a random factor. To account for experimental rounds, we added ‘round’ as a random factor. We checked for the necessity of random slopes and intercepts by the application of likelihood ratio tests of the full model with the effect in question against the model without the effect in question, finally resulting in a general random slope and intercept model for ‘Environment × Population’ (response ∼ treatment * pop + (treatment | pop/clone) + (1 | round)) and ‘Genotype × Environment’ (response ∼ treatment * clone + (1 | round)). Statistical significances for life history traits were obtained by using the function Anova (model, type = 2)) which performs a Wald Chi-Square test.

To assess shape variation we used the principal component analysis after the GPA in the R package ‘geomorph.’ Subsequently the statistical analysis was done with Procrustes ANOVA and pairwise tests to reveal statistically relevant ‘shape’ differences between groups.

Results

Effect of genotype origin on predator-induced response in life history traits: ‘Environment × Population’ effect

Reaction norms of life history traits per population (Fig. 1) as well as their boxplots (Fig. 2) show intra- and inter-population variation of life history traits. The statistical analysis revealed that the factor ‘Environment’ affected the life history traits ‘AFR’ and ‘broods’ in a significant manner. The factor ‘Population’ affected ‘offspring,’ ‘brood1,’ ‘relnest’ and ‘relclone,’ but there was no significant interaction effect of ‘Environment × Population’ for any life history trait (Table 2A). The visualization of growth differences between environments and populations differences of somatic growth rate (dSGR, Fig. 3) showed that all clonal lines from popG had a negative growth rate in the fish-exposed environment, resulting in a smaller body size. Four out of six clonal lines from popJ had a negative somatic growth rate, while clonal lines from popLC and popM vary in somatic growth rates across environments.

Figure 1 Reaction norms for selected life history traits showing population differences (mean +/− SE).

Population Greifensee (popG, yellow), population Jordan Reservoir (popJ, black), population Lake Constance (popLC, magenta) and population Müggelsee (popM, green). (A) Age at first reproduction (‘AFR’). (B) Total number of offspring first brood (‘brood1’). (C) Total number of broods (‘broods’). (D) Total number of offspring (‘offspring’). (E) Somatic growth rate (‘SGR’). (F) Body length (‘size’).

Figure 2 Boxplots for selected life history traits showing population differences (median +/− SD).

(A) Age at first reproduction (‘AFR’). (B) Total number of offspring first brood (‘brood1’). (C) Total number of broods (‘broods’). (D) Total number of offspring (‘offspring’). (E) Somatic growth rate (‘SGR’). (F) Body length (‘size’).

Table 2 Results of general linear mixed effect models (GLMM) on various life history traits.

(A) The effect of individual origin (‘Environment × Population’).	
		Environment × Population	
Life history trait		Chi-square	Df	Pr(>Chi-square)	
Age at first reproduction (‘AFR’)	Environment	5.0261	1	0.0250	
Population	3.5870	3	0.3097	
Environment × Population	1.9912	3	0.5742	
Total number of broods (‘broods’)	Environment	3.9718	1	0.0463	
Population	7.5636	3	0.0560	
Environment × Population	1.4309	3	0.6983	
Total number of offspring (‘offspring’)	Environment	1.5044	1	0.2200	
Population	17.1803	3	0.0006	
Environment × Population	1.9182	3	0.5860	
Total number of offspring first brood (‘brood1’)	Environment	0.0065	1	0.9358	
Population	10.3740	3	0.0156	
Environment × Population	0.6623	3	0.8031	
‘Survival’	Environment	0.2127	1	0.6447	
Population	0.2403	3	0.9708	
Environment × Population	0.0000	3	1	
Relative fitness within populations (‘relnest’)	Environment	0.2608	1	0.6096	
Population	9.8864	3	0.0196	
Environment × Population	0.1236	3	0.9889	
Relative fitness among populations (‘relclone’)	Environment	0.3751	1	0.5402	
Population	13.6158	3	0.0035	
Environment × Population	0.3250	3	0.9553	
Somatic growth rate (‘SGR’)	Environment	3.3442	1	0.0674	
Population	5.2943	3	0.1514	
Environment × Population	5.7855	3	0.1225	
Body length (‘size’)	Environment	3.5277	1	0.0604	
Population	2.1413	3	0.5436	
Environment × Population	5.2355	3	0.1553	
(B) The effect of ‘Genotype × Environment’ interaction.	
		Genotype × Environment	
Life history trait		Chi-square	Df	Pr(>Chi-square)	
Age at first reproduction (‘AFR’)	Environment	41	1	<0.001	
Genotype	253	23	<0.001	
Genotype × Environment	146	23	<0.001	
Total number of broods (‘broods’)	Environment	11	1	<0.001	
Genotype	114	23	<0.001	
Genotype × Environment	64	23	<0.001	
Total number of offspring (‘offspring’)	Environment	5	1	<0.001	
Genotype	988	23	<0.001	
Genotype × Environment	175	23	<0.001	
Total number of offspring first brood (‘brood1’)	Environment	0.0267	1	0.870	
Genotype	116	23	<0.001	
Genotype × Environment	34	23	0.060	
‘Survival’	Environment	3	1	0.0944	
Genotype	28	23	0.1991	
Genotype × Environment	8	23	0.9987	
Relative fitness within populations (‘relnest’)	Environment	2.6	1	0.1076	
Genotype	302	23	<0.001	
Genotype × Environment	NA	NA	NA	
Relative fitness among populations (‘relclone’)	Environment	2.6	1	0.1041	
Genotype	235	23	<0.001	
Genotype × Environment	NA	NA	NA	
Somatic growth rate (‘SGR’)	Environment	20	1	<0.001	
Genotype	1,289	23	<0.001	
Genotype × Environment	147	23	<0.001	
Body length (‘size’)	Environment	50	1	<0.001	
Genotype	1,613	23	<0.001	
Genotype × Environment	179	23	<0.001	
Note:

Significant values (p < 0.05) are highlighted in bold. Values are rounded.

Figure 3 Differences of somatic growth rate (dSGR).

Differences of somatic growth rate (dSGR) as μm per day (mean +/− SD); calculated as: mean of SGR (fish) minus mean SGR (control) equals dSGR per clonal line, sorted by populations. Each clonal line is displayed in its population specific colour. Population Greifensee (popG, yellow), population Jordan Reservoir (popJ, black), population Lake Constance (popLC, magenta) and population Müggelsee (popM, green).

The fittest population in control environment was popJ (w = 1), followed by popM (w = 0.83), popLC (w = 0.78) and popG (w = 0.67). In fish environment a small change of positions occurred for popLC and popM. Here the decreasing order was popJ (w = 1), followed by popLC (w = 0.80), popM (w = 0.77) and popG (w = 0.63) among all populations. Further details of relative fitness for each clonal line within their population can be found in Tables 3A and 3B.

Table 3 Relative fitness (w) within and among populations.

(A) Relative fitness within populations for genotype means.	
		w within population (‘relnest’)	w among populations (‘relclone’)	
Population	Clone	Control	Fish	Control	Fish	
G	G1.11	0.53	0.84	0.36	0.50	
G1.12	0.35	0.66	0.24	0.40	
G1.6	0.46	0.31	0.32	0.19	
G1.7	0.95	0.86	0.65	0.51	
G2.1	0.81	0.86	0.56	0.52	
G3.1	1.00	1.00	0.69	0.60	
J	J1	0.73	0.75	0.73	0.75	
J2	0.64	0.68	0.64	0.68	
J2.1	0.50	0.69	0.50	0.69	
J2.4	1.00	1.00	1.00	1.00	
J3	0.67	0.70	0.67	0.70	
J4	0.63	0.55	0.63	0.55	
LC	LC3.1	0.73	0.59	0.55	0.45	
LC3.3	0.56	0.63	0.42	0.47	
LC3.5	0.78	0.96	0.59	0.72	
LC3.6	1.00	1.00	0.75	0.75	
LC3.7	0.46	0.54	0.35	0.41	
LC3.9	0.78	0.95	0.59	0.71	
M	M10	0.72	0.97	0.43	0.66	
M12	0.87	0.71	0.52	0.48	
M2	0.98	0.86	0.59	0.59	
M5	0.95	1.00	0.57	0.69	
M6	0.82	0.88	0.50	0.60	
M9	1.00	0.78	0.60	0.54	
(B) Range of relative fitness among populations for genotype means.	
Population	w control	w fish	
G	0.24–0.69	0.19–0.60	
J	0.50–1.00	0.55–1.00	
LC	0.35–0.75	0.41–0.75	
M	0.43–0.60	0.54–0.69	
Note:

Fittest genotype or population (w = 1.0) is highlighted in bold.

Effect of ‘Genotype × Environment’ interaction on life history traits

In the model comprising the factors ‘Genotype’ and ‘Environment,’ most of the traits were significantly affected by both factors as well as their interaction: ‘AFR,’ ‘broods,’ ‘offspring,’ ‘SGR’ and ‘size’ (Table 2B). The traits ‘brood1,’ ‘relnest’ and ‘relclone’ were only affected by the factor ‘Genotype’ (Table 2B). Reaction norms for each life history trait of each clonal line can be found in the Supplementary Material (Figs. S2–S7).

Effect of fish kairomones on the morphological trait ‘shape’

A total of 83% of ‘shape’ variation was explained by the first four principal components (PC1 = 42%, PC2 = 24%, PC3 = 11% and PC4 = 6%) (Fig. S1).

The geometric morphometric analysis revealed a significant ‘Environment’ effect on ‘shape’ (Df = 455, F = 5.93, Pr(>)F = 0.001). Differences in shape occurred within each population except for popG (Table 4A) as well as among populations (Table 4B). There was a significant interaction effect of ‘Environment × Population’ on ‘shape’ (Df = 3, F = 1.9163, Pr(>)F = 0.019). The p-value matrix revealed a statistical significance difference within popLC between environments (p = 0.035; Table 4C). There was a statistical significant ‘Genotype × Environment’ effect on ‘shape’ (Df = 411, F = 3.2947, Pr(>)F = 0.047). Experimental ‘round’ was included in the geometric morphometric analysis revealing its statistical significant effects on the morphological trait ‘shape.’

Table 4 Results of geometric morphometric analysis.

(A) p-values of ‘Environment’ effect on ‘shape’ differences within populations.	
Population	Df	F	Pr(>F)	
G	1	0.39	0.843	
J	1	5.87	0.001	
LC	1	3.79	0.007	
M	1	3.16	0.009	
(B) p-value Matrix of ‘Environment’ effect on ‘shape’ among populations.	
–	G	J	LC	M	
G	–	0.013	0.167	0.013	
J	0.013	–	0.143	0.001	
LC	0.167	0.143	–	0.003	
M	0.013	0.001	0.003	–	
(C) p-value Matrix of the interaction of ‘Environment × Population’ on ‘shape.’	
–	G:control	G:fish	J:control	J:fish	LC:control	LC:fish	M:control	M:fish	
G:control	–	0.848	0.589	0.364	0.137	0.473	0.420	0.932	
G:fish	0.848	–	0.913	0.671	0.470	0.454	0.418	0.421	
J:control	0.589	0.913	–	0.534	0.731	0.352	0.629	0.614	
J:fish	0.364	0.671	0.534	–	0.172	0.290	0.892	0.290	
LC:control	0.137	0.470	0.731	0.172	–	0.035	0.103	0.248	
LC:fish	0.473	0.454	0.352	0.290	0.035	–	0.743	0.876	
M:control	0.420	0.418	0.629	0.892	0.103	0.743	–	0.399	
M:fish	0.932	0.421	0.614	0.290	0.248	0.876	0.399	–	
Note:

Statistical significant F-values (Pr(>F) < 0.05) are displayed in bold.

The shape of females with many offspring (n > 22 = upper quartile of total number of offspring) differed significantly among populations in the control environment, but not in the fish environment. There was no association between ‘shape’ and a high number of ‘offspring’ in the fish environment. Further analysis revealed that the ‘shape’ of females with many offspring did not differ significantly between environments within each population.

Visualization revealed an overall ‘shape’ change towards a smaller body. In detail, a homogenous change from all directions to a smaller body form was found for popG (Fig. 4B). Within popJ the overall shape change towards a smaller body size was shown with the strongest change in the head area (bending of the thin plate spline) and an anterior–posterior direction (Fig. 4C). Within popLC the head position changed from dorsal to ventral direction, while a small change of the tail area from a ventral to dorsal direction (Fig. 4D) occurred. Within popM the overall shape change towards a smaller body size was shown in the head area from a dorsal to ventral direction and in the tail area from a ventral to dorsal direction (Fig. 4E).

Figure 4 Thin plate spline (TPS) grids of consensus shapes of superimposed Procrustes coordinates.

Control (red). Fish (green). (A) All specimens. (B) Population Greifensee (popG). (C) Population Jordan Reservoir (popJ). (D) Population Lake Constance (popLC). (E) Population Müggelsee (popM).

Discussion

Intraspecific phenotypic variation of life history traits in D. galeata

Our study revealed a significant ‘Environment,’ ‘Genotype’ as well as ‘Genotype × Environment’ effect for the life history traits ‘AFR,’ ‘broods,’ ‘offspring,’ ‘SGR’ and ‘size.’ Concordant to previous studies (Boersma, Spaak & De Meester, 1998; Stibor & Lüning, 1994), our results showed a decrease of ‘AFR’, a decrease of ‘SGR’ and a decrease of body length in the presence of fish kairomones in D. galeata. Out of the 24 studied genotypes, 13 matured earlier (Fig. S2) and 17 reduced their body length (Fig. S7) in the presence of fish kairomones. Indeed, early maturation and a reduced size of Daphnia in the presence of vertebrate predators have been reported before (Machacek, 1991; Weider & Pijanowska, 1993; Lampert, 1993; Gliwicz & Boavida, 1996). The ecological benefit lies in a successful reproduction before reaching a body size making the individual vulnerable to fish predation (Lynch, 1980; Lampert, 1993).

The ‘Genotype × Environment’ effect for most of the life history traits implies that the presence or absence of certain clonal lines within one population might have an effect on overall population survival, depending on environmental factors such as predation risk. Hence, if the phenotypic diversity within one population is reduced and the majority produces relatively less offspring in a fish environment, the result could be an overall low number of offspring in the following cohorts, which would threaten the persistence of the whole population. Notably, individuals of popG produced less offspring and less broods compared to the other three populations regardless of the environment and their relative fitness was comparatively low. Potential explanations for this relative low performance of popG could be genetic drift and inbreeding depression which have a negative effect on genetic diversity (Vanoverbeke & De Meester, 2010). However, low genetic variation for D. galeata in Greifensee was not identified (Herrmann et al., 2017), making these two explanations unlikely at first glance. On the contrary, most clonal lines in Greifensee (four out of six) had a higher heterozygosity than expected, perhaps as a result of hybridization, which is known to occur in this population (Brede et al., 2009). Therefore, outbreeding depression could explain lower fitness in popG and should be further investigated in a future study.

Our experimental design allowed us to assess the distribution of variance at clonal as well as population level. We thus detected phenotypic variation within each as well as among several populations independent of the environment. We identified two different strategies of phenotypic plastic responses of D. galeata by comparing the environmental effect within as well as among the populations. In popJ, the variation of a trait itself, not the change in the trait median value as a response was extremely reduced for two life history traits, ‘AFR’ and total number of ‘broods’ (Fig. 2C). Almost all individuals of popJ started to reproduce at the same age and produce the same amount of broods in fish environment, showing a striking homogeneity under stress. On the contrary, in popM the variation for ‘AFR’ increased, resulting in a broader range of ‘AFR’ in fish environment. The phenotypic variation between clonal lines was best visualized by plotting the dSGR between the environments (Fig. 3), unifying the environmental and genotype effect. All six clonal lines of popG and four out of six clonal lines of popJ decreased their somatic growth in fish environment, while the direction of response varied for popLC and popM. Overall our study with a total of 24 clonal lines revealed a broad spectrum of phenotypic variation of life history traits in European D. galeata.

To our surprise we did not find an ‘Environment × Population’ effect on the life history response, although we observed intra- and inter-population differences, especially between the two extremes popG and popJ. A significant ‘Population’ effect was found for the interdependent traits total number of offspring (‘offspring’), size of first brood (‘brood1’), relative fitness within (‘relnest’) and among populations (‘relclone’). This observed significant population divergence could be due to the extreme difference of total number of offspring between popG and popJ. In general, genotypes in popJ produced the highest number of offspring among all populations. In contrast, the total number of offspring of genotypes in popG was overall lower compared to the other three populations, regardless of the environment. Even the increased number of offspring for clonal lines of popG in fish environment is lower than the numbers of offspring for clonal lines of popJ in control environment. Hence, the genotype origin (‘Population’) itself had little to no effect on life history traits in Daphnia implying that the identity of a clonal line (‘Genotype’) within a population seems to be more important than the origin of the clonal line per se.

In the end, we were not able to identify one main driving force influencing the phenotypic variation of life history traits in D. galeata, and could not infer a population specific response. Instead our study displays the complexity of the interacting factors ‘Environment’ and ‘Genotype’ to produce a variety of phenotypes within one species, thereby contributing to the understanding of intraspecific phenotypic variation.

Potential for local adaptation

Our findings allow the conclusion that there is a potential for local adaptation to predation risk in the investigated European populations of D. galeata. This conclusion was based on two outcomes of our study.

On one hand, we observed an extreme predator-induced life history response for popJ. The range of variation of the phenotypic response was reduced to a minimum in popJ, so that almost all individuals of the six clonal lines and 15 replicates reproduce at the very same age when exposed to fish (Fig. 2A). On top of that, we observed a similar reduction of variation for the life history trait total number of broods (Fig. 2C). These strong responses could be explained by local adaptation to the presence of fish. The Jordan Reservoir is an artificial inner city water reservoir, used for recreational purposes such as fishing since 1900 (Kubecka & Bohm, 1991) and had been regularly stocked with fish (Seda, Hejzlar & Kubecka, 2000). Therefore, D. galeata of Jordan Reservoir had the possibility to adapt to an environment with a higher predation risk for more than a century. Such micro-evolutionary changes for Daphnia species have been described in other contexts before. For instance, Jansen et al. (2011) showed that D. magna was able to evolve resistance to a pesticide (carbaryl) within experimental time. Further, Declerck, Cousyn & De Meester (2001) as well as Reger et al. (2018) showed that populations were able to locally adapt to fish kairomones (D. galeata and D. pulex, respectively). Alternatively, since the reservoir, unlike the other lakes in this study, has been created specifically with fishing in mind, differential colonization might also be the source of the observed pattern. This habitat might have been colonized only by Daphnia pre-adapted to fish, with very specific life-histories, leading to the present-day striking pattern.

On the other hand, the relative fitness of individuals of popJ suggests that females exposed to fish kairomones are fitter, concurring with results obtained by Castro, Consciência & Gonçalves (2007) and Jansen et al. (2011). Since local adaptation to a certain stressor implies a better performance in the stressful environment than without this stressor (Lenormand et al., 1999; Joshi et al., 2001) we suggest that the local adaptive potential exists for at least three populations because the relative fitness in the presence of fish kairomones increased overall for 14 out of 24 clonal lines (popG = 3, popJ = 4, popLC = 4, popM = 3) (Table 3A). Our results are in line with earlier studies showing the adaptive potential of phenotypic plasticity in Daphnia exposed to different stressors (Yin et al., 2011; Altshuler et al., 2011; Reger et al., 2018; Hesse et al., 2012).

Predation risk and morphological changes

In general, we did not observe any predator-induced extreme morphological changes such as the formation of helmets for fish kairomone exposed Daphnia as those reported for D. lumholtzi (Laforsch & Tollrian, 2004). We presented here the first study using the geometric morphometric analysis to measure morphometric changes to an environmental factor in D. galeata, hence complementing the traditional approaches (life history traits and behaviour). Our morphometric analysis revealed that the presence of fish kairomones had an effect on the body shape of D. galeata. However, no overall pattern was recognizable among the populations and no effect was observed at all for popG. Instead we observed different changes of ‘shape’ in each population. We suggest that the morphological trait ‘shape’ is phenotypically plastic due to high clonal variation, which is consistent with the results reported by Dlouhá et al. (2010) and Zuykova, Bochkarev & Katokhin (2012). The difference between experimental rounds for ‘shape’ could be attributed to this observed high clonal variation which we observed in all life history traits as well.

We hypothesized that life history change and morphological change are correlated, meaning that females with a higher number of offspring (n > 22, upper quartile of observed total number of offspring) would change their ‘shape’ towards a bulkier body form to accommodate a greater number of offspring within their brood pouch. This correlation was found only in individuals in control environment and not for individuals in fish environment. Changing shape of the body might come along with some drawbacks: the bulkier the shape, the higher the detection risk by the predator and the slower the swimming ability due to drag. In fact, fish prey size-selectively on Daphnia, meaning that larger Daphnia are preyed upon more often than smaller Daphnia (Weber & Van Noordwijk, 2002; Beckerman, Rodgers & Dennis, 2010). Since fish prey on faster swimming individuals of Daphnia (O’Keefe, Brewer & Dodson, 1998), being a slow swimming Daphnia would be beneficial. Alternatively, accommodating more offspring without changing the shape of the body might be achieved through the production of smaller offspring, as was shown by Castro, Consciência & Gonçalves (2007), among others (Lampert, 1993). In line with previous studies showing a predator-induced reduction in neonate size, we can speculate that this is also the case in our experiment and plan to further explore this dimension.

Conclusions

The study presented here focused on the assessment of intraspecific phenotypic variation in D. galeata. By comparing the range of phenotypic response of 24 clonal lines originating from four populations, we contribute to the understanding of the effect of environmental change (shifts in predator regime) on intraspecific phenotypic variation at the population level. We observed high clonal variation in all studied life history traits at the intra- and inter-population level, leading to the suggestion that single genotype studies on Daphnia might deliver biased conclusions.

Supplemental Information

Supplemental Information 1 Raw data table of life history traits for all experimental individuals.

This input file is needed to conduct the life history trait analysis part I (treatment effect) and part II (adaptive potential).

Click here for additional data file.

Supplemental Information 2 Input file for life history trait (LHT) analysis.

To calculate the relative fitness within and among populations this input file is needed for life history trait analysis part I and part II.

Click here for additional data file.

Supplemental Information 3 Input file for the visualization of dSGR.

This input file is needed to compute Figure 5 showing the differences of somatic growth rate (dSGR) per genotype.

Click here for additional data file.

Supplemental Information 4 Life history trait analysis part I.

This file includes code of the life history trait analysis (part I: ‘Genotype x Environment’ effect) for the R environment.

Click here for additional data file.

Supplemental Information 5 Life history trait analysis part II.

This file includes code of the life history trait analysis (part II: ‘EnvironmentxPopulation’ effect) for the R environment.

Click here for additional data file.

Supplemental Information 6 Visualizations of life history traits.

This file includes code of the visualizations of life history traits for the R environment.

Click here for additional data file.

Supplemental Information 7 Geometric morphometric analysis.

This file includes code of the geometric morphometric analysis for the R environment.

Click here for additional data file.

Supplemental Information 8 Information on experimental individuals.

This file includes information on experimental individuals needed for the geometric morphometric analysis in R.

Click here for additional data file.

Supplemental Information 9 Input file for geometric morphometric analysis of all specimen (control & experimental conditions).

TPS data of all specimen used for geometric morphometric analysis.

Click here for additional data file.

Supplemental Information 10 Input file for geometric morphometric analysis of all specimens exposed to experimental condition.

TPS data for all specimens exposed to fish kairomones.

Click here for additional data file.

Supplemental Information 11 Input file for geometric morphometric analysis of all specimens exposed to control conditions.

TPS data for all specimens exposed to control conditions.

Click here for additional data file.

Supplemental Information 12 Input file for geometric morphometric analysis of all specimens from population Greifensee (popG) exposed to control and experimental conditions.

TPS data for all specimens of popG exposed to control and experimental conditions.

Click here for additional data file.

Supplemental Information 13 Input file for geometric morphometric analysis of all specimens from population Greifensee (popG) exposed to fish kairomones.

TPS data of all specimens from popG exposed to fish kairomones.

Click here for additional data file.

Supplemental Information 14 Input file for geometric morphometric analysis of all specimens from population Greifensee (popG) exposed to control conditions.

TPS data of all specimens of popG exposed to control conditions.

Click here for additional data file.

Supplemental Information 15 Input file for geometric morphometric analysis of all specimens from population Jordan Reservoir (popJ) exposed to control and experimental conditions.

TPS data of all specimens from popJ exposed to control and experimental conditions.

Click here for additional data file.

Supplemental Information 16 Input file for geometric morphometric analysis of all specimens from population Jordan reservoir (popJ).

TPS data for geometric morphometric analysis of all specimens from popJ exposed to fish kairomones.

Click here for additional data file.

Supplemental Information 17 Input file for geometric morphometric analysis of all specimens from population Jordan reservoir exposed to control conditions.

TPS data of all specimens from popJ exposed to control conditions.

Click here for additional data file.

Supplemental Information 18 Input file for geometric morphometric analysis of all specimens from population Lake Constance (popLC) exposed to control an environmental conditions.

TPS data of all specimens from popLC exposed to control and experimental conditions.

Click here for additional data file.

Supplemental Information 19 Input file for geometric morphometric analysis of all specimens from population Lake Constance (popLC) exposed fish kairomones.

TPS data of all specimens from popLC exposed to fish kairomones.

Click here for additional data file.

Supplemental Information 20 Input file for geometric morphometric analysis of all specimens from population Lake Constance (popLC) exposed to control conditions.

TPS data of all specimens from popLC exposed to control conditions.

Click here for additional data file.

Supplemental Information 21 Input file for geometric morphometric analysis of all specimens from population Müggelsee (popM) exposed to control an environmental conditions.

TPS data of all specimens from popM exposed to control and environmental conditions.

Click here for additional data file.

Supplemental Information 22 Input file for geometric morphometric analysis of all specimens from population Müggelsee (popM) exposed to fish kairomones.

TPS data of all specimens from popM exposed to fish kairomones.

Click here for additional data file.

Supplemental Information 23 Input file for geometric morphometric analysis of all specimens from population Müggelsee (popM) exposed to control conditions.

TPS data of specimens from popM exposed to control conditions.

Click here for additional data file.

Supplemental Information 24 Table S1. Overview of all clonal lines used in experimental rounds.

Click here for additional data file.

Supplemental Information 25 Fig. S1. Principal component (PC) plot of overall shape variation.

PC plot of superimposed Procrustes coordinates of all specimen. The thin plate spine grids show shapes associated with the positive end of the horizontal axis and the negative end of the vertical axis.

Click here for additional data file.

Supplemental Information 26 Fig. S2. Reaction norms for the life history trait age at first reproduction (‘AFR’).

Genotype mean (+/−SE) within one population are displayed for the trait ‘AFR’ in days. The overall within population mean (+/−SE) is displayed in a population specific color. (A) Population Greifensee= popG= ’yellow.’ (B) Population Jordan Reservoir= popJ= ’black.’ (C) Population Lake Constance= popLC= ’magenta.’ (D) Population Müggelsee= popM= ’green.’.

Click here for additional data file.

Supplemental Information 27 Fig. S3. Reaction norms for the life history trait total number of broods (‘broods’).

Genotype mean (+/−SE) within one population are displayed for the trait ‘broods’. The overall within population mean (+/−SE) is displayed in a population specific color. (A) Population Greifensee= popG= ’yellow’. (B) Population Jordan Reservoir= popJ= ’black’. (C) Population Lake Constance= popLC= ’magenta’. (D) Population Müggelsee= popM= ’green’.

Click here for additional data file.

Supplemental Information 28 Fig. S4. Reaction norms for the life history trait total number of offspring (‘offspring’).

Genotype mean (+/−SE) within one population are displayed for the trait ‘offspring’. The overall within population mean (+/−SE) is displayed in a population specific color. (A) Population Greifensee= popG= ’yellow’. (B) Population Jordan Reservoir= popJ= ’black’. (C) Population Lake Constance= popLC= ’magenta’. (D) Population Müggelsee= popM= ’green’.

Click here for additional data file.

Supplemental Information 29 Fig. S5. Reaction norms for the life history trait total number of offspring first brood (‘brood1’).

Genotype mean (+/−SE) within one population are displayed for the trait ‘brood1’. The overall within population mean (+/−SE) is displayed in a population specific color. (A) Population Greifensee= popG= ’yellow’. (B) Population Jordan Reservoir= popJ= ’black’. (C) Population Lake Constance= popLC= ’magenta’. (D) Population Müggelsee= popM= ’green’.

Click here for additional data file.

Supplemental Information 30 Fig. S6. Reaction norms for the life history trait somatic growth rate (‘SGR’).

Genotype mean (+/−SE) within one population are displayed for the trait ‘SGR’ in μm/day. The overall within population mean (+/−SE) is displayed in a population specific color. (A) Population Greifensee= popG= ’yellow’. (B) Population Jordan Reservoir= popJ= ’black’. (C) Population Lake Constance= popLC= ’magenta’. (D) Population Müggelsee= popM= ’green’.

Click here for additional data file.

Supplemental Information 31 Fig. S7. Reaction norms for the life history trait body length (‘size’).

Genotype mean (+/−SE) within one population are displayed for the trait ‘size’ in μm. The overall within population mean (+/−SE) is displayed in a population specific color. (A) Population Greifensee= popG= ’yellow’. (B) Population Jordan Reservoir= popJ= ’black’. (C) Population Lake Constance= popLC= ’magenta’. (D) Population Müggelsee= popM= ’green’.

Click here for additional data file.

We thank Jens Oldeland, Bob O’Hara and Suda Parimala Ravindran for valuable statistical advice. Additionally, we thank Michael Engelmohn, Tatjana Usinger and Anne Ehring for their help during Daphnia breeding and the experiment. We would like to thank Lisa Gottschlich for testing and confirming geometric morphometric measurements and Thomas Mehner for his input on fish density calculation. Earlier versions of this manuscript greatly benefited from the comments of three anonymous reviewers.

Additional Information and Declarations

Competing Interests

Author Contributions

Animal Ethics

Data Availability

The authors declare that they have no competing interests.

Verena Tams conceived and designed the experiments, performed the experiments, analysed the data, prepared figures and/or tables, authored or reviewed drafts of the paper, approved the final draft.

Jennifer Lüneburg performed the experiments, approved the final draft.

Laura Seddar performed the experiments, approved the final draft.

Jan-Phillip Detampel analysed the data, prepared figures and/or tables, approved the final draft.

Mathilde Cordellier conceived and designed the experiments, contributed reagents/materials/analysis tools, authored or reviewed drafts of the paper, approved the final draft.

The following information was supplied relating to ethical approvals (i.e., approving body and any reference numbers):

Animal handling and experiments were in accordance with the ethical standards (approved for the execution of experiments on vertebrates No: 75/15).

The following information was supplied regarding data availability:

The raw data are provided in the Supplemental Files.

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
