# Peer review of "Intraspecific phenotypic variation in life history traits of Daphnia galeata populations in response to fish kairomones"

_PeerJ, doi:10.7717/peerj.5746_

## Round 0.1 · original submission · Major Revisions

You have received three very thorough reviews of your manuscript, and I would ask that you carefully consider the reviewers' comments and indicate your point-by-point response to each in an accompanying letter submitted together with your revised manuscript.

Reviewer 1 ·

Basic reporting

This is an experimentally sound paper and will be of interest to readers because it explores the effects of population and clone on responses to predation. I commend the authors for taking up this topic—it is important to include more than 1-2 genotypes in studies such as these!
Please see below for my general comments. I think the Results and Discussion should be streamlined and reorganized so that the readers can more easily follow along. The authors also should more clearly report their statistical methods in the Methods so that the results can be more easily interpreted.


Good job in Intro with reporting background, though I do think that it would be useful to use more specific examples (see General Comments). The literature is well referenced and relevant.

In general, the language is clear, but the paper should be reviewed by native English speaker. E.g., Abstract line 18-19 isn’t structured properly, word choice in lines 362 (should be notably) and 394 (unfolds isn’t correct).

Figures:
Figure 1: put axes
Figure 3: could be supplemental
Figure 4: Why are the data for the treatment shown as dots and data for the random effects shown as bars? The legend description is also confusing (first paragraph).
Figure 5: put axes
Table 3: why is there a matrix of p-values? Is this the results of pairwise posthocs?

Raw data are supplied.

Experimental design

Within scope of journal.

Well-defined research question and clear statement of its usefulness. You are right that exploring one genotype within a population is problematic.

Line 108-109: curious what happens when you use “time since lineage established” as covariate. This is just a suggestion, I don't think it's necessary.

Line 200-201: what were your treatments and random factors? In general, you need more detailed description of actual models so that reader can properly evaluate your stats.

I cannot fully evaluate the validity of the findings because the statistical methods are not fully described. It seems like a glm should be used for all analyses, with treatment, population and clone as fixed effects and block as a random effect.

Validity of the findings

Difficult to assess validity because statistical models not reported. See above.

Conclusions need to be more clearly stated. In general, the organization of the Results and Discussion make it hard to keep track of the patterns. You have a lot of results to present, so organization is key. As stated throughout, I recommend organizing these sections by response variable rather than treatment.

However, I agree with the authors’ interpretations that there is inter-population and clonal variability.

Additional comments

Abstract line 12 – not sure conditions is the right verb
Introduction
Line 40-41: Plasticity is favored in variable environments and is important because it helps organisms in heterogeneous environments. It would be worth saying that here to distinguish it from general phenotypic variation.
Line 51: why not just call them kairomones instead of infochemicals?
Line 51: helpful to discuss the specific predator induced defenses (e.g. morphological defenses or behavioral shifts) rather than just saying that predation triggers phenotypic changes.
Line 73: how are these changes adaptive? E.g. smaller size would not seem to be adaptive at first glance.
Methods
Line 103: is “large lakes” an ecotype? I am not a Daphnia researcher, so this may be something that people in your field recognize, but if not, it should be clarified what this means.
Line 127: good job standardizing the number of generations
Line 131: I am confused about which individuals you are talking about for the remainder of this paragraph: the breeding and experimental individuals?
Line 214-217: would be better to give effect sizes rather than raw changes
Line 218-223: why aren’t p-values given for population and clone effects? Were population and clone treated as fixed factors?
Results, general comments: I find the organization of the methods confusing, but this could be because the models weren’t described in the Methods. I assume the fixed factors and their interactions were analyzed in the same statistical model. Why then are they divided into separate sections? I suggest re-organizing by response variable rather than fixed effect, for example, like your description of body shape.
Line 258-261: This interpretation should be saved for the Discussion.
Line 278: There was no description of the PCA in your Methods.
Discussion
Line 323: be more specific than “different organizational levels”
Line 326-327: Do you mean the mean of the trait vs the variance of the trait?
Line 340-342: worth briefly summarizing here even if briefly discussed before.
Line 344: be specific about why more bacteria.
I like how the “driving forces of phenotypic variation” section is organized. It is easy to understand. I recommend replicating this throughout the Discussion.
Line 351: Why was genotype a random factor if this is one of the main factors you are examining in this experiment?
Line 391: Confused by the word “variation” because it implies differences in diversity. Do you mean the identity of the genotype is more important?
Line 397: I like this section of the Discussion. It is well written and gives a good overview of the Discussion so that the reader has general take aways. Consider replicating throughout Discussion.
Line 451: If being slow is beneficial, why would Daphnia in the fish treatment not be bigger?

Reviewer 2 ·

Basic reporting

The manuscript presents data on the responses of differed Dapnhia clonal lineages from 4 different populations to predation risk (chemical cues). Although it is always interesting to see these kind of experiments showing variability in plastic responses to predation risk I feel that the study lacks some focus. It is well known that Daphnia clones exhibit variance in response to chemical stressors also to biotic stressors like predator kairomones.
In this sense is not clear how the authors are integrating results with their hypothesis. Also a better review on the relevant literature on this would help.

Experimental design

The results of this study come from laboratory tests and I have some questions about the experimental design and analysis of data that should be addressed before publication. Some justification of the chosen experimental design is also lacking. See comments below

Validity of the findings

The results show variability in the responses of different clones but I feel that they lack contextualization since no information is provided concerning the different predation pressures of the origin sites of the different clones. Moreover, some effect of the block design somewhat hampers the validity of findings. I also think that the discussion section lacks depth. See comments below

Additional comments

Although the manuscript is in general well written authors should make an effort to include some more relevant literature on this especially concerning all the plastic responses that have been observed in Daphnia exposed to predator chemical cues. In fact, responses such as feeding inhibition or increased metabolic costs have been also described for Daphnids exposed to fish kairomones and should be referenced. However, authors should integrate their results better. As such it is well known that There are genetic and population differences in the plasticity and in the magnitude of induced defences. The specific response of a population to predator’s chemical cues is influenced by the evolutionary history with respect to the presence of a particular predator in a particular habitat. Therefore, populations which co-occur with predators show local adaptation to habitat-specific predation regimes and stronger responses towards predators than naïve prey. In this sense it would be critical for authors to present some info of the different predation pressures from the different lakes?
At the same time, it is well known that different genotypes can also respond differently to the presence of predators, and this difference relies not on the presence/absence of a response but also on different combinations and magnitudes of the different traits showing that different responses can be uncoupled and evolve independently.
The discussion section and the important role of phenotypic plasticity and evolutionary potential of daphnid populations would have greatly benefited from an in-depth discussion on these important issues. Also a better justification and discussion of the following:

1- Authors state that “The Jordan Reservoir is an artificial inner city water reservoir,
used for recreational purposes such as fishing since 1900 (Kubecka and Bohm 1991) and had been regularly stocked with fish (Seda et al 2000). Therefore, D. galeata of Jordan reservoir had the possibility to adapt to a high-predator environment for more than a century.” This is all fine but without some information concerning the predation pressure of the other lakes we cannot conclude that popJ was “better adapted” to fish predation pressure.
2- Also on this why was Leuciscus idus used as a model? Any evidences that this fish species exists in any of the lakes where these populations were taken from? This is important again since different Daphnid responses have been shown towards kairomones of different fish species.
3- Since Responses to fish predation risk is stronger when alarm cues from conspecifics are present and if the objective was to show plasticity and variability is response in a number of different traits why weren’t they used?
4- Why didn’t authors prepared a single stock kairomones solution to be used throughout experiments? This would be the only way to guarantee that the kairomones concentration is the same and would eventually allow for testing all clones simultaneously. In fact, the significant block effect is hard to neglect.
5- I don’t understand why the different clonal lineages were maintained in control medium AND in kairomones solution for 2 generations prior to testing. If the induced responses are plastic, all clones should have been maintained in control medium (so as to remove any maternal effects) and only exposed to kairomones in the actual test. If I understood correctly the life history data reported correspond to a 2 generation exposure. Do authors think this might have affected the results?
6- Also responses to fish kairomones are known to be concentration dependent, why did authors decide to use only one kairomones treatment? I think that using a lower kairomones concentration would have helped in comparing the magnitude of responses of the different clones.
7- Why did authors addressed fitness without taking into consideration Age at first reproduction. This parameter is critical isn the response to predation pressure ( in natural conditions with predators present Daphnids don’t usually get to the third , fourth broods so easily…). A better Proxy for fitness with these kind of experiments might be the calculation of R (intrinsic rate of population increase. Which is influenced greatly by age at first reproduction and size of first brood
8- I am not sure if the number of genotypes within each population is sufficient but wouldn’t these results be better analysed using quantitative genetic analyses?
By analyzing the norms of reaction of the different clones, different micro-evolutionary paths can be analysed by correlation measures of fitness traits across environments. Maybe mean values and variances for each trait can be calculated and genetic variation of a trait among treatments (environments) will be compared using the genotypic coefficient of variation (CVG). Wouldn’t this be better to evaluate the evolutionary potential of each population?
9- Authors state that males were eliminated from analyses. Shouldn’t the whole replicate be eliminated? Did those genotypes produced males in control medium?
10- I think that the investigation of morphological responses interesting but it is strange to see that the size of neonates within the different treatments was not presented. This is important for the discussion (L451-455). Also measuring size at maturity could also give important clues.
11- Authors state that ( L 357-362) “The main effect of ‘Genotype’ on the traits offspring and SGR implies that the presence or absence of genotypes within one population might have an effect on overall population survival. If the clonal diversity within one population is reduced and the majority of genotypes which produce relatively less offspring are present in a predator environment, the result could be of an overall low number of offspring in the following cohorts which would threaten the persistence of the whole population.” I don’t undertand this and it should be rephrased and discussed with the plastic responses and the chemical cues that induced them in mind. In principle these plastic responses are adaptive so we can expect that the genotypes showing more plastic responses (reproducing earlier at a smaller size and producing more and smaller neonates would be better adapted in an environment of higher fish predation pressure. As it is the sentence is too vague, too confusing.
12- Finally, authors should be coherent throughout the manuscript and refer predation risk or fish kairomones when referring the results of their experiments and not predation or predator presence since no actual predation was tested nor were other cues ( visual , hydrodynamic) . (e.g. L21-22) . Also avoid the term vertebrate kairomones (L397). Again be careful when using the term “maturity” or early maturation since you didn’t assess age or size at maturity.

Reviewer 3 ·

Basic reporting

The introduction section is in need of focus and improved clarity. The information I would like to see in the introduction is:
1. What is phenotypic plasticity (a definition – maybe something involving an interaction between genotype and the environment). Why is phenotypic plasticity important for evolution? (as opposed to selection on existing phenotypic variation)
2. What do we know about how predators can influence life history traits and morphometric traits – how much is due to selection vs. plasticity? (do we know?)
3. What is the main question the researchers are addressing that leads them to the hypotheses they make in the final intro paragraph

Related to these ideas, I have these thoughts:
- The first paragraph is confusing. What is the main point the authors want to make? This is where it will be important to both define phenotypic plasticity and why it is important (from an ecological/ evolutionary standpoint) to know more about it. It feels backward to begin by talking about the importance of plasticity without first explaining what it is.
- Lines 67-75: I’m not quite sure what main points the authors want to make. Line 72-73 is strange – plasticity is not necessary for adaptation. But maybe the issue is the meaning of “adaptation”. My expectation is that adaptation means evolution, which does not need to involve plasticity. Though plasticity may certainly evolve. I think the authors should explain what they mean by adaptation and why plasticity is necessary for adaptation to take place. Lines 73-75 are also not very informative.
- Lines 76-87: This paragraph should define the research question inspiring your work. Right now, it’s pretty vague, about comparing clones in different populations and the major focus is on replicate numbers. That’s great that this study has greater replication at the clone level than other past studies; however, six clones still isn’t a lot to base an entire population on. What question(s) are you addressing? Beyond having more clones per population than anyone else? What specifically are you interested in examining? Why do you want to look at within population and across population variation in phenotypic plasticity? This paragraph should lead naturally to the next one where you describe your hypotheses/predictions, but right now the predictions come out of nowhere because there’s no research question.

Experimental design

This article needs some work in defining the research question and in describing the methods in sufficient detail.

There are some important details that are missing in the methods section.
1) Line 105: These clones come from sediment cores from the four lakes. Were these sediment cores dated? Do you know the time span of the sediment slice that the clones come from? This is important for understanding the variation observed for each population. If the clones span 25 years from one lake but 4 years for another, this could be an issue to consider.
2) What do you know about variation in predation pressure among lakes and over time in the lakes? Why do you expect to see variation in predation responses among these populations? This important ecological aspect is barely addressed in the study. The first mention of the predation environment in any of the lakes is line 409 and I don’t see any mention of the other three lakes. I find this very perplexing. How can you study GXE interactions with so little discussion of the environment that the populations come from?
3) Lines 199-206: What were the actual statistical analyses performed to test the various hypotheses? As is, the statistical methods are completely insufficient to understand the analyses that were performed and then interpret the results and assess the discussion section.
Smaller methodological details
4) Line 115-116 – Why do you starve the fish to avoid the Daphnia receiving alarm cues from the kairomones? Isn’t that the point of the study, to make the Daphnia feel the threat of predation and respond to it?
5) Line 167-169 – Were the Daphnia measured alive or preserved in ethanol?

Validity of the findings

I am unable to assess the results and conclusions section because there is so little information available in the methods on what was done (statistical methods in particular). I also want to know about the predation regime in the lakes to know how to interpret the variation seen in plastic responses to fish kairomones.

Based on what I understand from the results tables, one question I have is about why so many measures of fecundity were included as response variables. Wouldn’t there be an issue of these variables being highly correlated with one another (e.g. size of brood, number of broods, total number of offspring)?

Additional comments

The authors have undertaken an impressively large study. They have rigorously measured fitness and morphological traits of four populations of Daphnia galeata in a control and predator cued environment. The major issue I see with this manuscript is lots of detail in some areas (e.g. figures and results) but little attention to the big picture (e.g. important background information, what is plasticity and why does it matter for evolution, what ecological settings do the four populations experience, why did the authors measure what they measured). I also see an important lack of detail in key aspects of the methods (what statistical tests were performed??). While I think this study could be very interesting, at present there are some major issues that need to be addressed.

---

## Round 0.2 · Major Revisions

The revised version of your manuscript was returned to two of the three original reviewers. As you can see, one of these reviewers has recommended that the revised manuscript be accepted for publication, whereas Reviewer 3 recommends rejection. I have decided to give you one more opportunity to address the critical reviewer's concerns and revise your manuscript accordingly. Should you decide to revise your manuscript, please provide a point-by-point document indicating how you have responded to each of the reviewer's concerns.

Reviewer 2 ·

Basic reporting

The manuscript presents data on the responses of different Dapnhia clonal lineages from 4 different populations to predation risk (chemical cues).

The authors have reviewed the manuscript addressing most of mine (and other reviewers ) comments.

Experimental design

The results of this study come from laboratory tests and authors have satisfactorily replied to my questions/comments. Most details missing are due to the size and workload necessary, which I fully understand and other analysis are being prepared for different papers. Details of fish presence in the different collection sites are now provided.

Validity of the findings

I continue to feel that effect of the block design somewhat weakens authors findings but the statistical analysis section was improved and these results deserve to be published

Additional comments

I agree with other reviewers that this study is important to show variability within Daphnid populations towards predation risk. The results are interesting and deserved to be published. Authors did a good job reviewing the manuscript and correcting some details while adding the necessary information on methods and statistical analysis. Main concerns/ comments and suggestions made by reviewers were addressed in this version which I think now meets the standards for publication in PeerJ.

Since authors were not certain of having found the right studies concerning metabolic / feeding responses of daphnids to fish predation risk I leave some suggestions of papers that can be cited :

Rose, R. M., Warne, M. S. J., and Lim, R. P. (2003). Exposure to chemicals exuded by fish reduces the filtration and ingestion rates of Ceriodaphnia cf. dubia. Hydrobiologia 501, 215–217. doi:10.1023/A:1026284314506

Beckerman, A. P., Wieski, K., and Baird, D. J. (2007). Behavioural versus physiological mediation of life history under predation risk. Oecologia 152(2), 335–343. doi:10.1007/S00442-006-0642-6

Pestana JLT, Baird DJ, Soares AMVM (2013) Predator threat assessment in Daphnia magna: the role of kairomones versus conspecific alarm cues. Mar Freshwater Res 64:679. doi: 10.1071/MF13043

Pijanowska, J. (1997). Alarm signals in Daphnia? Oecologia 112(1), 12–16. doi:10.1007/S004420050277

Reviewer 3 ·

Basic reporting

The overall structure (introduction, methods, results, discussion) is appropriate, but I feel this article needs a good deal of revision to be ready for publication. The lack of clarity, abundant detail in some cases and lack of detail in others hampered my ability to read, understand, and evaluate this manuscript.

Experimental design

I am not totally sure the experimental design is appropriate now that I know more about the study. The nature of the predation regime in the study lakes is a really important thing to understand here.

I still see very limited information about the predation regimes in the four study lakes. I believe the new information is in the supplemental table S1. There is a row showing the number of kg/ha of fish in each lake. This tells us very little about the predation regime from the perspective of the daphniids in the lakes. When were the fish surveys conducted? At the same time of year for each lake? What kinds of fish are these? Are they planktivorous? Also, are any of these the type of fish used for the kairomone assays?

If you want to know why you see variation in predation kairomone responses, it seems to me that the predation regime that the population experiences is important for interpreting that variation. Based on the response to Reviewer 2's comments It sounds like the fish used to produce kairomones is a model species but maybe not a species that lives in any of the lakes? I would find this problematic if this is true. But this should also be made clear in the text of the paper.

I am disappointed that a study focusing on the plasticity of predation responses in natural populations gives so little attention to describing or trying to understand the selection pressures in the lakes. How can you or the reader hope to interpret the variation that you observe?

In addition, I still do not know the time periods when the Daphnia were hatched. The authors mention they’re from the same 5 year time span in each lake. What time span is that? Does it correspond with when the fish densities were calculated? Just citing the studies that used these cores previously is not enough information. Also, from having done paleolimnological work in multiple lakes myself it seems kind of unlikely that there was a sediment slice for each lake that was deposited during the exact same 5 year time span. Is this really what the authors mean?

Validity of the findings

It s still a bit difficult for me to assess the findings since I don’t know enough about the environmental pressures these Daphnia populations experience.

I am also concerned with the way the authors have discussed the block effect. If the block effect is significant I think you just need to take that variation into account while seeing if other variables are significant. You can’t just ignore the fact that it’s significant. I totally understand why blocking needs to be done, but you can’t ignore a significant block effect (lines 411-413)

Additional comments

I appreciate the authors providing some additional information in response to the reviewers’ comments. However, I am really struggling to read through the manuscript and understand what the authors have found. I think the main issue is that a lot of details are reported but without a coherent structure or story. There are also important things that are not shared in enough depth (e.g. predation regime).

My recommendation is to have several colleagues provide feedback to help shape this manuscript and make it easier to understand and have a clear message. It is just too much work right now to figure out what the authors are trying to get across and I cannot do my job as a reviewer to assess the strength of the science. I would also suggest focusing on the results that support your key finding rather than listing all the results in the order they were entered into the models.

I recently learned of this blog post that gives some great tips for scientific writing. You might like to take a look! https://dynamicecology.wordpress.com/2016/02/24/the-5-pivotal-paragraphs-in-a-paper/

---

## Round 0.3 · accepted · Accept

Thank you for your thorough responses, particularly to Reviewer 3's concerns. I believe your revisions have made the message much clearer.

#